# Prediabetes-Associated Changes in Skeletal Muscle Function and Their Possible Links with Diabetes: A Literature Review

**DOI:** 10.3390/ijms25010469

**Published:** 2023-12-29

**Authors:** Mandlakazi Dlamini, Andile Khathi

**Affiliations:** Department of Human Physiology, School of Laboratory Medicine and Medical Sciences, College of Health Sciences, University of KwaZulu-Natal, Durban X54001, South Africa; 218004674@stu.ukzn.ac.za

**Keywords:** type 2 diabetes mellitus, prediabetes, skeletal muscle, satellite cells, myogenic regulatory factors, insulin resistance, muscle fibers, inflammation, oxidative stress

## Abstract

The skeletal muscle plays a critical role in regulating systemic blood glucose homeostasis. Impaired skeletal muscle glucose homeostasis associated with type 2 diabetes mellitus (T2DM) has been observed to significantly affect the whole-body glucose homeostasis, thereby resulting in other diabetic complications. T2DM does not only affect skeletal muscle glucose homeostasis, but it also affects skeletal muscle structure and functional capacity. Given that T2DM is a global health burden, there is an urgent need to develop therapeutic medical therapies that will aid in the management of T2DM. Prediabetes (PreDM) is a prominent risk factor of T2DM that usually goes unnoticed in many individuals as it is an asymptomatic condition. Hence, research on PreDM is essential because establishing diabetic biomarkers during the prediabetic state would aid in preventing the development of T2DM, as PreDM is a reversible condition if it is detected in the early stages. The literature predominantly documents the changes in skeletal muscle during T2DM, but the changes in skeletal muscle during prediabetes are not well elucidated. In this review, we seek to review the existing literature on PreDM- and T2DM-associated changes in skeletal muscle function.

## 1. Introduction

The skeletal muscle is one of the most prominent insulin-sensitive tissues in the body and functions as the primary site for insulin-stimulated glucose uptake [1]. Alterations in skeletal muscle health can affect whole-body glucose homeostasis as the skeletal muscle, in conjunction with the liver, is chiefly involved in regulating glucose uptake and maintaining glucose homeostasis [2,3]. Skeletal muscle satellite cells are among the most paramount progenitor cells responsible for maintaining skeletal muscle health under physiological and pathophysiological conditions [2]. Satellite cells play a critical role in muscle fiber maintenance, repair, and remodeling, ultimately maintaining skeletal muscle plasticity [4]. Chronic metabolic diseases, such as diabetes mellitus, have been observed to affect skeletal muscle health by negatively modulating satellite cell quantity or functionality [2].

Diabetes mellitus is a metabolic disorder characterized by chronically elevated blood glucose levels due to defective insulin release or function [2]. Approximately 422 million people have been diagnosed with diabetes mellitus globally, with the majority living in low- and middle-income countries [5]. Type 2 diabetes mellitus (T2DM) is the most prevalent type, accounting for approximately 90% of global diabetes cases [2,6]. T2DM is anticipated to affect almost 8% of the worldwide population by 2030 [7]. T2DM is characterized by insulin resistance, where the body cells cannot effectively respond to insulin action, which leads to hyperglycemia. Unhealthy lifestyle behaviors, such as sedentary lifestyle combined with chronic consumption of high caloric diets, result in the onset of impaired glucose tolerance and insulin resistance seen in T2DM [8]. Prediabetes (PreDM) is characterized by blood glucose levels higher than those in the homeostatic range, but below the threshold for diabetes mellitus for a diagnosis of T2DM [9]. It is observed that both fasting glucose levels and glucose tolerance are impaired during the prediabetic state [9]. T2DM has been observed to considerably compromise skeletal muscle health, a phenomenon known as diabetic myopathy [2]. Diabetic myopathy is associated with reduced physical capacity, strength, and muscle mass [10,11,12]. Diabetic myopathy is one of the understudied complications of diabetes mellitus, and it is proposed to be directly involved in the rate of comorbidity development [2]. 

However, the onset of T2DM is often preceded by an asymptomatic condition known as PreDM [9]. There are several studies that have suggested that the onset of complications associated with T2DM begin during the prediabetic state. The literature predominantly documents the changes in skeletal muscle during T2DM, but the changes in skeletal muscle during PreDM are not well elucidated. In this review, we seek to review the existing literature on PreDM- and T2DM-associated changes in skeletal muscle function. The following section describes the role of skeletal muscle satellite cells in skeletal muscle health maintenance.

## 2. Role of Satellite Cells in Skeletal Muscle

Skeletal muscle can adapt to various stimuli via the modulation of muscle size, fiber-type distribution, and metabolism [2]. This phenomenon is due to the skeletal muscle progenitor cells, particularly satellite cells, playing a role in skeletal muscle maintenance and plasticity [2]. Skeletal muscle satellite cells are vital for skeletal muscle fiber maintenance, repair, and remodeling [4]. Satellite cells are generally latent in adult skeletal muscle and become only functional upon stimulation. Stimulation of satellite cells results in satellite cell activation, proliferation, and differentiation [4]. Myoblasts, the progeny of satellite cells, play a role in skeletal muscle growth and the regeneration of satellite cells. Skeletal muscle growth mediated by myoblasts occurs by the combination of myoblasts to form new myofibers or combine with an existing muscle fiber and donate their nucleus during the fusion process. Satellite cell regeneration occurs when myoblasts return to a quiescent state, which replenishes the resident pool of satellite cells [13]. 

Paired box transcription factor 7 (Pax7) and myogenic regulatory factors (MRFs), such as MyoD, Myf5, MRF4, and myogenin, are observed to regulate the function of satellite cells during myogenesis [4]. Pax7 is mainly expressed in quiescent satellite cells and plays a role in self-renewal and maintenance of the basal satellite pool [14]. Studies illustrate that there is a functional overlap between the MRFs in establishing myogenesis. MyoD and Myf5 are proposed to induce myoblast activation and proliferation, whereas myogenin and MRF4 are proposed to induce the terminal differentiation of satellite cells [15]. In a study of newborn mice lacking MyoD and Myf5, it was observed that the mice were devoid of myoblasts and myofibers [16]. Contrarily, mice with myogenin deficiency generated myoblasts but demonstrated insufficient skeletal muscle differentiation, with minimum and smaller myotubes [17,18]. Hence, the coordinated action of MRFs is vital for establishing myogenic lineage and terminal myogenic phenotype [19]. MRFs consist of a basic helix-loop-helix (bHLH) domain that enables them to recognize and bind to the E-box sequence (CANNTG) (Figure 1). The heterodimerization of MRFs with a ubiquitously expressed E-protein family of bHLH proteins (i.e., E12) orchestrates the binding of MRFs to the E-box sequence (Figure 1) [20,21].

Studies document that subpopulations of satellite cells can undergo asymmetric divisions to synthesize myogenic progenitors or symmetric divisions to increase the satellite cell pool [22]. Moreover, satellite cells are observed to also commit to the myogenic lineage and proliferate to give rise to committed myogenic progenitors, which can asymmetrically divide or directly differentiate into myocytes that will fuse and form new myofibers [22] (Figure 2). The ability of satellite cells to be able to choose between performing asymmetric or symmetric divisions enables them to coordinate their activity with the needs of the regenerating muscle. The increased propensity of symmetric division during muscle regeneration would stimulate the expansion of the satellite cell pool [23] (Figure 2). In contrast, the asymmetric division would favor the generation of myogenic progenitors and maintenance of the stem cell pool (Figure 2). Thus, a dynamic balance must be established between the fluctuating symmetric and asymmetric divisions that occur during the different stages of muscle regeneration, as an imbalance would result in muscle regeneration impairment [22]. Hence, MRFs can be used as a biomarker to assess satellite cell function in myogenesis. The following sections will outline PreDM- and T2DM-associated changes in skeletal muscle function.

## 3. Prediabetes

PreDM, a prominent risk factor of T2DM, is characterized as an asymptomatic condition, as individuals rarely present with physical symptoms, such as polydipsia, polyphagia, and fatigue, as seen in T2DM [5,24]. However, the literature documents that there are some microvascular and macrovascular changes that begin in the prediabetic state that may contribute to the long-term complications of T2DM if prediabetes progresses to T2DM [9,24,25]. A large proportion of the global population is predisposed to prediabetes. The global prevalence rate of prediabetes in 2017 was estimated to be 352,1 million (7.3%) of the adult population, and it is anticipated to increase to 587 million (8.3%) individuals by 2045 [26]. Prediabetes is characterized by higher than normal blood glucose levels but not high enough to establish a T2DM diagnosis and presents as a risk for the onset of T2DM [9]. According to the WHO [5], the prediabetes diagnostic criteria include individuals presenting with one or both of impaired fasting glucose (IFG) or impaired glucose tolerance (IGT). IFG is characterized by fasting plasma glucose (FPG) concentration ≥6.1 mmol/L and <7 mmol/L and IGT is characterized by FPG concentration <6.1 mmol/L and a 2-h post-load plasma glucose concentration between ≥7.8 mmol/L and <11.1 mmol/L measured during the oral glucose tolerance test (OGTT) [27]. Glycated hemoglobin A1c (HbA1c) levels between 5.7 and 6.4% are also used for prediabetes diagnosis [27]. Several factors such as genetic predisposition, insulin resistance, glucotoxicity, lipotoxicity, and β-cell dysfunction result in prediabetes development [9].

Studies document prediabetes to be related to early forms of micro- and macrovascular diabetic complications such as nephropathy, chronic kidney disease, small-fiber neuropathy, diabetic retinopathy, and heightened risk of macrovascular disease [28]. Studies have observed an increased risk of coronary disease during the prediabetic state [29,30]. Considering that the onset of T2DM complications occurs during the prediabetic state, prediabetic conditions need to be well elucidated as this would aid in preventing some of the overlapping prediabetic and T2D complications.

### 3.1. Effects of PreDM on Skeletal Muscle Glucose Homeostasis

The insulin resistance associated with PreDM is documented to contribute to the onset of endothelial dysfunction [31,32,33] (Figure 3). Insulin is vital for endothelial function and glucose metabolism [34]. Studies document that insulin induces vasodilation in resistance arterioles, increases compliance of large arteries, promotes capillary recruitment, and maintains capillary permeability to support nutrient delivery [35,36]. The effects of insulin require coordinated downstream events to keep the vascular tone in the basal state while the vasodilatory response to insulin is elevated in the postprandial state. The skeletal muscle microvasculature, therefore, links insulin’s vascular and metabolic functions by increasing the surface area for tissue perfusion. Considering that the skeletal muscle is the principal tissue for insulin-stimulated glucose disposal, these insulin actions represent the role of the endothelium in regulating glucose homeostasis [34]. 

Insulin stimulates the production of nitric oxide (NO) from the endothelium, promoting vasodilation, increased blood flow, and enhanced glucose disposal in skeletal muscle [37]. The insulin-dependent production of NO in vascular endothelium is facilitated by the insulin receptor (IR) tyrosine kinase that phosphorylates insulin receptor substrate 1 (IRS), resulting in the binding and activation of phosphoinositide 3 (PI 3)-kinase and phosphoinositide-dependent protein kinase (PDK-1). PDK-1 phosphorylates and activates Akt and also phosphorylates and activates eNOS. The subsequent increase in NO generation stimulates vasodilation and increases blood flow. Insulin-mediated glucose uptake in skeletal muscle involves a similar signaling pathway culminating in the translocation of GLUT4 glucose transport to the cell surface [38]. In healthy individuals, insulin’s vasodilator actions promote insulin’s direct effects on glucose transport in skeletal muscle and adipose tissue, resulting in increased glucose uptake [38]. In metabolic and cardiovascular disease, including diabetes, obesity, and atherosclerosis, inflammatory signaling through the inhibitor of nuclear factor-kβ kinase (IKKβ) in response to cytokines and elevated free fatty acids (FFA) levels induces insulin resistance in both vascular endothelium and metabolic targets of insulin, such as the skeletal muscle [38,39] (Figure 4). Thus, the inflammatory mechanisms of insulin resistance impact vascular endothelium and metabolic targets of insulin similarly, contributing to both metabolic and cardiovascular diseases [38,39] (Figure 4).

Increased FFAs induce insulin resistance and endothelial dysfunction in obese patients with prediabetes [39,40] (Figure 4). A high-fat diet triggers endothelial dysfunction in mice [41] and eating a meal high in fat reduces brachial artery reactivity in humans [42]. FFAs decrease tyrosine phosphorylation of IRS-1/2 and inhibit the PI3K/Akt pathway, resulting in reduced glucose transport and reduced phosphorylation of eNOS [43,44,45]. FFAs activate NADPH oxidase via protein kinase C (PKC) to generate reactive oxygen species (ROS) [46]. Activated PKC contributes to endothelial permeability [47] and extracellular matrix (ECM) expansion [48]. 

Elevated oxidative stress is associated with the activation of several serine/threonine kinases and the activation of transcription factors NF-kB and activator protein (AP-1), which result in insulin resistance [49]. The activation of serine/threonine kinases c-Jun NH_2_-terminal kinase (JNK), PKCs, and IkB kinase complex β (IKKβ) leads to serine phosphorylation of IRS-1, which disrupts its ability to bind and activate PI 3-kinase. Thus, there is reduced activation of downstream kinases Akt and PKC-ζ, which reduces the translocation of GLUT4 and glucose transport [50,51,52] (Figure 4). The skeletal muscle plays a substantial role in glucose homeostasis [49]. Hence, more studies must be conducted to contribute to the current understanding of prediabetic complications within the context of skeletal muscle. 

### 3.2. Effects of PreDM on Skeletal Muscle Vasculature and Extracellular Matrix

The pro-inflammatory prediabetic state promotes increased skeletal muscle collagens and other ECM proteins [53], including fibronectin, proteoglycans, and connective tissue growth factors, and ECM remodeling [54]. The glycosaminoglycans hyaluronan is elevated in tissues of insulin-resistant animals [55,56]. Hyaluronan is a prominent component of the glycocalyx of capillary lumens, which may affect insulin access to tissues [40]. Reduced hyaluronan using PEGylated hyaluronidase in high-fat-fed mice ameliorates insulin action [55]. Expansion of the muscle ECM and decreased muscle capillary are proposed to contribute to muscle insulin resistance [57].

The chronic systematic inflammation associated with a high-fat diet [58,59] is suggested to heighten ECM protein synthesis and decrease ECM degradation, leading to increased deposition and ECM remodeling [60,61] (Figure 5). The increased protein expression within the ECM is hypothesized to induce a physical barrier, impeding normal insulin action and glucose diffusion across the sarcolemma [55,62] (Figure 5). 

The increased protein expression is suggested to be associated with collagen, fibronectin, and proteoglycan proteins, which accumulate in the interstitial space, resulting in increased diffusion distance and prevention of substrate and hormonal delivery [54,62]. In support of this hypothesis, Kang et al. [55] illustrated that hyaluronan (a significant ECM component) in skeletal muscle was remarkably increased in insulin-resistant diet-induced obese (DIO) mice when compared to normal chow-fed mice. Interestingly, the same authors also demonstrated that treatment with long-acting pegylated human recombinant PH20 hyaluronidase (PEGPH20) induced a dose-dependent decrease in muscle hyaluronan content and improved skeletal muscle insulin resistance in DIO mice [55]. These results suggest that depletion of ECM polysaccharide promotes muscle insulin sensitivity in obese mice, and contrarily, ECM protein accumulation seems to aggravate muscle insulin resistance [55].

Another hypothesized factor that is suggested to be linked to the underlying mechanism of ECM-associated insulin resistance in DIO is that the muscle ECM may expand to disrupt vascular function and neovascular growth, provided there is close contact between the ECM and endothelium [62]. Nutrient delivery to the contracting muscle requires functional blood flow to establish sufficient glucose (during exercise) and insulin (post-exercise) availability to facilitate glucose uptake and glycogen resynthesis, respectively [61]. Hence, ECM vascular dysfunction and capillary rarefaction have been associated with insulin resistance and T2D [62,63]. Studies show that 40% of insulin-stimulated glucose uptake is attributed to augmented muscle perfusion; however, in the insulin-resistant state, this hemodynamic response is observed to be absent [64,65,66]. Insulin-resistant models and humans are documented to present with capillary rarefaction, thereby highlighting the importance of sufficient muscle capillarization for insulin-mediated disposal [64,67]. 

Knockout mice lacking vascular endothelial growth factor (*VEGF*^−/−^) are proposed to have reduced insulin-mediated glucose disposal [64]. Importantly, Bonner et al. discovered that the reduction in skeletal muscle insulin-mediated glucose uptake was not associated with a dysregulation in intracellular insulin signaling (IRS-1, p85, and phosphorylated total (p/t) Akt), proposing that reduced insulin-stimulated muscle glucose uptake was caused by inadequate muscle perfusion [64]. Consequently, it has been suggested that it is challenging to elucidate insulin signaling in skeletal muscle as it is possible that integrin-associated signaling could have been implicated in *VEGF*^−/−^ rodent models [64]. The studies above suggest that ECM remodeling contributes to skeletal muscle insulin resistance via endothelial dysfunction and capillary rarefaction [62]. 

### 3.3. Effects of PreDM on Skeletal Muscle Satellite Cells

Prediabetes is observed to be associated with mitochondrial dysfunction, which is one of the suggested mechanisms related to reduced skeletal muscle satellite cell differentiation and insulin resistance [68,69,70]. Reduced mitochondrial biogenesis, reduced mitochondrial content, a low protein content, and decreased electron transport chain (ETC) complexes result in mitochondrial dysfunction. Such modifications potentially induce decreased substrate oxidation, such as lipids [71] (Figure 6A). The subsequent reduction in lipid oxidation, particularly the FFA, results in lipid accumulation, including deposition of metabolically active lipid mediators such as diacylglycerols (DAG) and ceramides (CER) [71,72] (Figure 6C). DAG contributes to triacylglycerol (TAG) synthesis and plays a role in activating protein kinase C (PKC). Activation of PKC by DAG promotes the translocation of PKC to plasma membrane and inhibits the insulin receptor and the subsequent phosphorylation of the insulin receptor substrate (IRS), ultimately resulting in decreased insulin action [71,72] (Figure 6C). Contrarily, CER inhibits insulin signaling via the inhibition of phosphatidylinositol-3 kinase (PI3K), which results in the inactivation of Akt, and subsequent reduced insulin action [73,74,75] (Figure 6C). A decreased electron flow through the ETC results in electron leakage and superoxide generation, subsequently inducing oxidative stress and damage. The superoxide and other ROS generated by oxidative stress cause mitochondrial damage, potentially resulting in either mitophagy (removal of damaged mitochondria and cell death prevention) or apoptosis. The removal of mitochondria via mitophagy could reduce mitochondrial quantity, leading to decreased substrate oxidation, further aggravating lipid accumulation and subsequent insulin resistance [71] (Figure 6B).

Insulin resistance induces impaired fasting glucose levels or glucose tolerance, hyperinsulinemia, and elevated FFA concentrations, inhibiting myoblast differentiation [76,77]. Lipids are documented as regulators of satellite cell function [78,79]. Skeletal muscle lipoprotein lipase (LPL) facilitates the entry of lipids into the muscle by catalyzing the hydrolysis of TAGs found in very-low-density lipoproteins or chylomicrons circulating in the bloodstream at the surface of the capillary endothelium [80]. The TAGs are hydrolyzed into free fatty acids, which are taken up by the muscle parenchyma cells [80]. Lipid overload (resulting from LPL overexpression) in the muscle is suggested to inhibit early myogenesis, impair satellite cell proliferation, and reduce regeneration capacity [81]. Myogenesis dysregulation is proposed to be associated with proteasomal activity, apoptosis, and skeletal muscle damage induced by ectopic lipid accumulation [81]. Latent satellite cells predominantly use fatty acid and pyruvate oxidation, and activated satellite cells utilize aerobic glycolysis [82]. The asymmetric division of satellite cells coincides with mitochondrial biogenesis [83]. The differentiation of satellite cells is associated with increased mitochondrial biogenesis, promoting oxidative phosphorylation [84,85]. Functional autophagy programs are required to facilitate the metabolic adjustments that occur during each phase of myogenesis [86]. Hence, negative modulation of autophagy may affect satellite cell function and skeletal muscle repair [87].

Xu et al. [70] documented the inhibition of skeletal muscle differentiation in a palmitic acid-induced insulin resistance model and an increase in glucose uptake and insulin sensitivity upon enhancement of skeletal muscle cell differentiation. In another study, inhibition of C2C12 myoblast differentiation was observed in a palmitic acid-induced insulin resistance model, whereas glucose uptake activity and insulin sensitivity gradually increased during C2C12 myoblast differentiation [88]. Thus, these findings confirm that lipotoxicity-related prediabetes may inhibit myoblast differentiation [70]. However, there are few studies in the literature documenting the prediabetic-associated changes in skeletal muscle satellite cells [70]. Hence, more studies are required to explore the changes associated with prediabetes in skeletal muscle satellite cells, as the literature documents that satellite cells play a pivotal role in skeletal muscle function [70]. 

### 3.4. Effects of PreDM on Skeletal Muscle Strength

Studies indicate that there is an association between prediabetes and reduced appendicular muscle strength [89,90]. The diminished muscle strength is attributed to insulin resistance, elevated inflammatory cytokines, and advanced glycosylation end products (AGEs) associated with prediabetes and type 2 diabetes [91,92]. The impaired insulin action associated with prediabetes is proposed to contribute to the dysregulation of protein synthesis, resulting in decreased muscle mass and strength [93]. The hyperglycemia that is associated with prediabetes is suggested to result in the accumulation of AGEs in the skeletal muscle. AGEs are documented to be associated with reduced grip strength, leg extension strength, and walking speed [94]. The increased inflammatory cytokines during the prediabetic state, suggested to be induced by skeletal muscle fat accumulation, are documented to be associated with reduced muscle mass and strength [95]. 

Studies illustrate that the association between increasing blood glucose levels, weaker muscle strength, and physical dysfunction is more prevalent in prediabetic and T2D males than females [90,96]. The pathogenesis of these findings is not well established [90]. A possible explanation behind these findings is related to lower intramuscular triglyceride oxidation and turnover rate in men, with the subsequent accumulation of intramuscular triglycerides [97]. This metabolic difference between genders may account for the lower muscle quality and physical performance mainly observed in prediabetic men [97]. Prediabetic men have also been observed to have lower insulin sensitivity than women, potentially due to the lower oxidative capacity and/or synthesis of the DAG pool [97]. Thus, alterations in intramuscular lipid metabolism likely occurs later in diabetes development in women than in men [90]. To date, there are few studies documented in the literature that explore the prediabetic skeletal muscle strength-associated changes; hence, more studies need to be conducted to address the mechanisms underlying the differential effects of glycemic abnormalities on physical function [90].

## 4. Type 2 Diabetes Mellitus

Type 2 diabetes is one of the leading types of DM, accounting for approximately 90% of global DM cases [2]. T2DM is associated with peripheral insulin resistance, impaired regulation of hepatic glucose production, and decreased pancreatic β-cell function, eventually leading to β-cell failure [98]. An oral glucose tolerance test (OGTT), a fasting glucose test, a postprandial glucose test, or glycated hemoglobin (HbA1c) test can be used to diagnose T2DM [98]. Type 2 diabetes is established when fasting blood glucose (FBG) levels are ≥7 mmol/L, postprandial glucose concentration is ≥11.1 mmol, and glycated hemoglobin concentrations are ≥6.5% [99]. Other organs implicated in T2DM development, other than the pancreas, include the liver, skeletal muscle, kidneys, brain, small intestine, and adipose tissue [100]. T2DM results in micro- and macrovascular complications such as nephropathy, neuropathy, cardiovascular diseases, and diabetic myopathy [2]. Diabetic myopathy is associated with reduced physical capacity, strength, and muscle mass [12]. Diabetic myopathy is proposed to be directly involved in the rate of comorbidity development; however, it is a relatively understudied diabetic complication [2].

### 4.1. T2DM-Associated Complication: Diabetic Myopathy

The pathophysiology of diabetic myopathy or diabetic muscle infarction (DMI) is not well understood; however, it is suggested to be a secondary manifestation of a microvascular pathological process associated with long-term poorly controlled DM, resulting in inflammation, ischemia, and infarction of the compromised muscles [101,102,103]. One of the potential mechanisms related to diabetic myopathy is the thromboembolic events elicited by microvascular endothelial damage and ischemic necrosis. As a result, there is impaired endothelium-dependent dilation in arterioles, elevated oxygen radicals, and diminished nitric oxide following reperfusion of ischemic tissues. The imbalance between oxygen radicals and nitric oxide in endothelial cells generates and secretes inflammatory mediators (tumor necrosis factor and platelet-activating factor) [103,104]. The inflammatory cascade heightens intracompartmental ischemia from edema, exacerbating tissue necrosis [105]. Studies also document DMI-associated modifications in the coagulation-fibrinolysis system, manifesting as hypercoagulability and vascular endothelial damage [103,106].

The muscle groups usually affected include muscles of the lower extremities, such as the thigh and the calf muscles [101,102], presenting as acute pain and swelling [103,107]. The upper limb extremities are seldom affected [105,108]. Over the years, diabetic myopathy has been more frequently documented in type 1 diabetic (T1D) patients than in T2D patients [102,109]. A possible explanation for this difference is that preceding cases of diabetic myopathy in T2D patients may have been unreported or misdiagnosed. Another possible reason is that the T2D patients may eventually require insulin for better glycemic control, resulting in these patients being reported as “insulin-dependent diabetic” and mislabeled as T1D patients [101]. Magnetic resonance imaging (MRI) is one of the most widely used diagnostic tools of DMI. Typical MRI parameters include a hyperintense signal on T2-weighted images and an isointense to hypointense signal on T1-weighted images from the affected muscle, with associated perifascial, perimuscular, and/or subcutaneous edema [103,110,111]. Studies document that MRI scans of 103 cases of T1D and T2D DMI depicted edema with T2 hyperintensity identified in 76.8% of cases, while T1 isointensity or hypointensity was reported in 14.6% of cases [103]. The initial presentation of thigh pain or swelling has been documented in 83.7% [107], 75% [112], and 80% [102] of DMI case [103]. Calf pain or swelling was recorded to be the second most common presentation, identified in 19, 28% [107], and 15% [112] of DMI [103]. 

### 4.2. Effects of T2DM on Skeletal Muscle Strength

Diabetic myopathy is documented to be associated with reduced muscle mass and appendicular muscle strength [113,114]. Andersen et al. [115] observed T2D-associated muscle strength loss around the ankle and knee joints. Studies document a positive correlation between reduced thigh muscle strength, muscle ectopic fat deposition, and insulin resistance in T2D patients [114]. Furthermore, age in T2D patients was associated with muscle ectopic fat and decreased muscle strength [114]. The fat infiltration in the muscle associated with muscle ectopic fat deposition, combined with low mitochondrial oxidative capacity, is suggested to potentially distort muscle architecture, resulting in the loss of muscle strength [116,117]. The knee isokinetic muscle strength measurement is used to assess flexion and extension muscle strength of the lower extremity, and the hand grip strength test is used to evaluate the muscle strength of the upper extremity [118]. 

Previous studies on the relationship between handgrip strength and T2DM have been conflicting [119]. Some studies document a significant inverse association between handgrip strength and T2DM [120,121,122,123] and some studies observed no significant association between handgrip strength and T2DM [124,125]. Studies illustrate that handgrip strength differs between ethnic groups, possibly accounting for the conflicting findings [123]. The low grip strength associated with T2DM is observed to be substantially higher in the South Asian population than in the Western population [123]. The inflammation associated with T2DM is also proposed to be related to low muscular strength [95]. In another study, high tumor necrosis factor-alpha (TNF-α) levels were associated with a decline in muscle strength [126]. The loss of skeletal muscle mass caused by T2DM results in decreased surface area for glucose transport, further exacerbating insulin resistance [127].

Studies document that T2D individuals have reduced type I muscle fibers and elevated type IIx muscle fiber proportions, potentially accounting for the reduced functional capacity observed in T2D individuals [128,129]. Type I levels directly correlate with insulin sensitivity; hence, the reduced expression of type I muscle fibers associated with T2DM is suggested to contribute to skeletal muscle insulin resistance [130,131,132]. Studies document a significant shift from slow oxidative fibers (type I) to fast glycolytic fibers (type IIa, IIb, IIx) in T2D individuals, which is observed to be associated with reduced oxidative enzyme activity [133] and increased glycolytic metabolism [134]. Muscle creatine kinase (MCK) synthesis is observed to be reduced in T2D rats, which is proposed to possibly result from the loss of MEF-1 binding activity observed in T2D conditions [135]. Furthermore, myosin heavy chain (MHC) IIb synthesis is reduced in the gastrocnemius muscle of T2D rats [135]. Myosin light chain (MLC) 1 and 3 isoforms are also observed to be decreased in T2D rats [135]. The reduction in skeletal muscle fiber MHC and MLC isoforms is associated with diminished muscle fiber number and size, subsequently impacting muscle mass and muscle strength [20]. 

### 4.3. Effect of T2DM on Skeletal Muscle Satellite Cells

T2DM has been observed to alter the function of satellite cells involved in muscle growth and regeneration [2]. Satellite cell function has been proposed to be considerably affected by hyperglycemic and lipotoxic conditions associated with type 2 diabetic states. For instance, a study discovered that three weeks of high-fat feeding (HFF) affected satellite cell content and functionality. The latter was characterized as the quantity of regenerating fibers present following injury [136]. Another study documented reduced muscle regeneration following eight months of HFF, which was proposed to be induced by delayed myofiber maturation [137]. A study conducted in the obese Zucker rat model for metabolic syndrome documented reduced satellite cell proliferative capacity; however, quiescent species remained unaltered [138]. A similar outcome was reported in another T2D study, whereby SC cell proliferation and activation were compromised, thereby affecting muscle regeneration [139].

Oxidative stress associated with T2DM is proposed to impair myogenesis [2]. Myogenesis is regulated by an integrated interaction between myogenic regulatory factors (MRFs), such as Myo, Jun D, and myogenin. These MRFs specifically bind to the muscle enhancer factor (MEF)-1 site, which regulates gene transcription of light and heavy chains of myosin and myosin creatine kinase (MCK) [140,141,142]. Disruption in the coordinated interaction between MRFs and the MEF-1 site can affect muscle protein synthesis and subsequently compromise skeletal muscle health [135]. Studies illustrate reduced myogenic factors (MyoD, myogenin, and Jun D) in STZ-diabetic and Zucker diabetic rodents [135]. MEF-1 DNA binding activity is also observed to be altered in T2D rats [135]. Myosin creatine kinase and myosin expression are also observed to be impaired due to reduced MEF-1 DNA binding activity associated with T2DM [143]. The binding of the homo- and heterodimers of MFRs to the MEF-1 site tightly regulates the development and differentiation of skeletal muscle progenitor cells into multinucleated myotubes [144]. 

### 4.4. Effects of T2DM on Skeletal Muscle Extracellular Matrix

The skeletal muscle extracellular matrix (ECM) consists of several proteoglycans and fibrous proteins such as collagens, fibronectin, laminins, and elastins, which contributes to the malleability of the skeletal muscle ECM [145,146]. Since the skeletal muscle ECM is highly malleable, its texture and, consequently, physiological roles may be affected by exercise, aging, or various diseases, such as diabetes [146]. ECM remodeling and variations in the expression patterns of integrin receptors are observed to influence the maintenance of satellite cell niche and myogenesis [147]. Skeletal muscle ECM remodeling is reported to be linked to diet-induced insulin resistance [54]. Studies show that insulin-resistant skeletal muscle has increased collagen deposition in both human and rodent models [54]. T2D-induced hyperglycemia upregulates fibronectin, laminin, and collagen, which can promote thickening of the basement membrane and diabetes-associated microangiopathy [148]. Integrins, crucial ECM molecule receptors, influence insulin activity regulation [62]. The skeletal muscle consists of seven α integrin subunits (α1, α3, α4, α5, α6, α7, αv) and a β1 integrin subunit [62]. The deficiency of integrin β1 subunit in mice’s skeletal muscle has been associated with insulin resistance. The insulin resistance was measured by reduced insulin-mediated glucose uptake and glycogen synthesis and indicated by diminished phosphorylation of protein kinase B (AKT) Ser-473 [149]. Studies document increased Collagen I and III in T2D and non-T2D obese individuals, with overfeeding being associated with higher expression of genes linked to intramuscular connective tissue (IMCT), and changes in intracellular pathways related to ECM receptor [150]. IMCT, integrins, and matrix metalloproteinases (MMPs) have been depicted to be involved in the progression of insulin resistance, particularly in overfeeding conditions [57].

MMPs are proteolytic enzymes responsible for regulating the homeostasis of myofiber functional integrity via degrading ECM proteins and regulating migration, differentiation, and regeneration of satellite cells [151]. There is low MMP activity under normal conditions, but it increases during recovery, disease-associated remodeling, or in inflamed tissue [151]. The skeletal muscle only has a few MMPs, such as MMP-2 and MMP-9, which are known to be part of muscle repair and remodeling. MMP-2 and MMP-9 play a substantial role in the regenerative process, as they are significantly expressed after skeletal muscle damage and assist in facilitating specific functions related to muscle repair and remodeling [152]. MMPs initiate degradation of ECM components, primarily collagens, resulting in satellite cell migration, proliferation, and differentiation of wounded or diseased skeletal muscle [151]. Since MMPs are responsible for the depletion of all ECM components, their dysregulation is also linked to the pathology of diabetes and obesity [153]. MMP-9 activity in skeletal muscle of high-fat fed mice is associated with decreased MMP-9 activity, which is indirectly related to the deposition of muscle collagen, but directly related to skeletal muscle insulin resistance [57].

## 5. PreDM-Associated Changes in Skeletal Muscle and Their Mechanistic Links to T2DM

Skeletal muscle endothelial dysfunction, mitochondrial dysfunction, and ECM remodeling are suggested to contribute to the insulin resistance that is observed in the onset of PreDM and T2DM [40,62,63,71,154]. The endothelial dysfunction is suggested to be induced by PreDM pathophysiological factors such as lipid accumulation, insulin resistance, and impaired insulin secretion [154]. Skeletal muscle intracellular lipid accumulation is proposed to dysregulate skeletal muscle insulin-mediated glucose uptake, which is controlled by vascular endothelial growth factor (VEGF)-B-stimulated lipid transport across the endothelium [154,155]. The upregulated VEGF-B-stimulated skeletal muscle lipid uptake observed in the prediabetic state is also documented in the T2D state [156]. Skeletal muscle VEGF-B binds to VEGF receptor 1 (VEGFR1) on the endothelial cells to heighten the expression of fatty acid transport proteins (FATPs). The FATPs result in increased lipid transport across the endothelium to the skeletal muscle, where it builds up as lipid droplets in the skeletal muscle, inducing insulin resistance and increasing blood glucose and pancreatic β cell death [156]. Ectopic lipid deposition in the skeletal muscle is associated with the pathogenesis of T2DM and PreDM [40,156]. Studies document that targeting the VEGF-B promotes insulin sensitivity and prevents the pathogenesis of T2DM by reducing lipid accumulation in skeletal muscle [157].

Studies illustrate that ECM remodeling is a common characteristic of insulin resistance in the skeletal muscle tissue, as seen in the prediabetic and T2D states [54,153]. The increased skeletal muscle ECM component synthesis is one of the significant features of prolonged T2D complications and is observed during the prediabetic state [153]. Elevated depositions of skeletal muscle ECM components, such as collagens and hyaluronan, along with ECM receptor integrins and CD44, are reported to contribute to PreDM- and T2DM-associated skeletal muscle insulin resistance [158]. The pro-inflammatory state associated with prediabetes is suggested to be associated with increased ECM components, which result in ECM remodeling [53]. Individual components of the remodeled ECM, such as collagen, laminin, and fibronectin, are documented to induce insulin resistance because the accumulation of ECM results in the formation of a physical barrier of the ECM and thus increases the diffusion distance for nutrients and insulin in the muscle [159].

The mitochondrial dysfunction associated with prediabetes is suggested to induce oxidative stress and insulin resistance, hallmark features of T2DM [71]. Mitochondrial dysfunction is associated with impaired mitochondrial oxidative capacity, resulting in increased intramyocellular fatty acids availability, which may contribute towards lipotoxic lipid species biosynthesis, such as ceramide and diacylglycerol, both of which have been associated with insulin resistance [160]. Elevated FFA acid concentrations in the skeletal muscle related to PreDM are suggested to induce oxidative stress via increased mitochondrial reactive oxygen species (ROS) production, which can directly cause skeletal muscle insulin resistance and trigger oxidative damage to mitochondrial DNA, proteins, and lipids, promoting the removal of damaged mitochondria by mitophagy (Table 1) [160]. Thus, oxidative stress is documented to interfere with the insulin signal transduction pathway, directly promote insulin resistance [161], and indirectly impede insulin signaling via mitochondrial damage and mitophagy [160]. The consequent decrease in mitochondrial function and density affects the overall skeletal muscle cellular oxidative capacity, ultimately favoring the ectopic accumulation of lipotoxic lipid intermediates, contributing to insulin resistance observed in the prediabetic and T2D state [160].

## 6. Conclusions

The research discussed in this review suggests that the skeletal muscle plays a substantial role in glucose homeostasis and illustrates the mechanisms involved in the onset of insulin resistance in the prediabetic and T2D states. PreDM is suggested to be related to early forms of T2D complications such as diabetic myopathy. A plethora of factors influence skeletal muscle function, among which are the satellite cells of skeletal muscles. Skeletal muscle satellite cells are proposed to maintain skeletal muscle health in physiological and pathophysiological conditions. Hence, the mechanisms involved in regulating skeletal muscle satellite cells’ effective functioning need to be well elucidated in the prediabetic state, as the literature mainly documents that satellite cell function is potentially impacted during the T2D state. Diabetic myopathy is a condition that occurs in long-term, poorly controlled T2DM. However, understanding the changes that take place in the skeletal muscle during the progression of PreDM can help us to be able to target and prevent the processes that could potentially contribute to the onset of diabetic myopathy in poorly controlled or uncontrolled T2DM. 

## Figures and Tables

**Figure 1 ijms-25-00469-f001:**
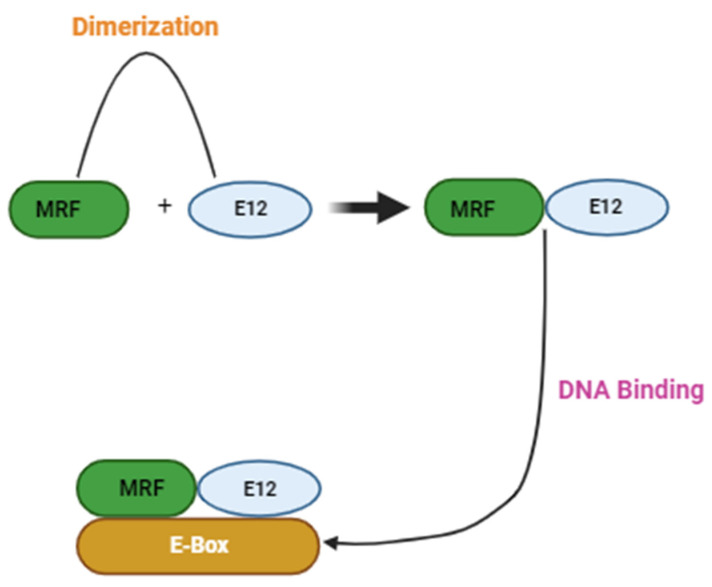
Schematic representation of myogenic regulatory factor (MRF) DNA binding activity, adapted from Torres-Machorro et al., 2021 [21].

**Figure 2 ijms-25-00469-f002:**
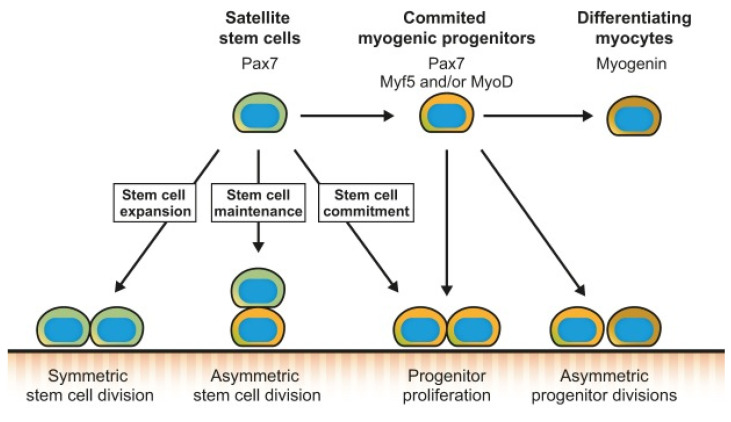
Schematic representation of symmetric and asymmetric satellite cell divisions, adapted from Dumont et al., 2015 [22].

**Figure 3 ijms-25-00469-f003:**
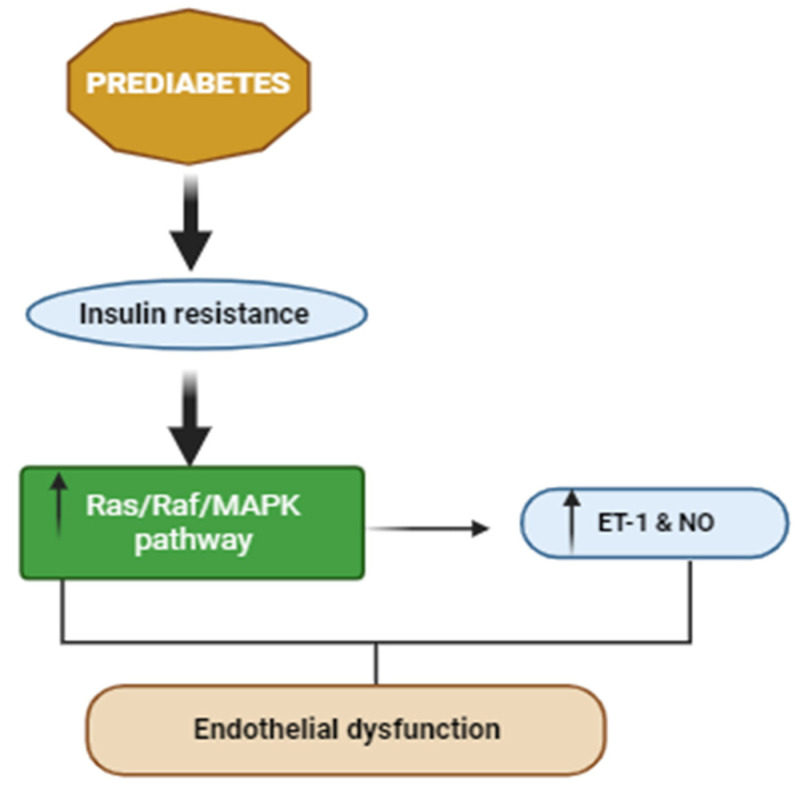
PreDM-associated modifications leading to endothelial dysfunction, adapted from Matteo Ciccone et al., 2014 [33]. MAPK: mitogen-activated protein kinase; ET-1: endothelin-1; NO: nitric oxide. ↑ = increase/upregulation.

**Figure 4 ijms-25-00469-f004:**
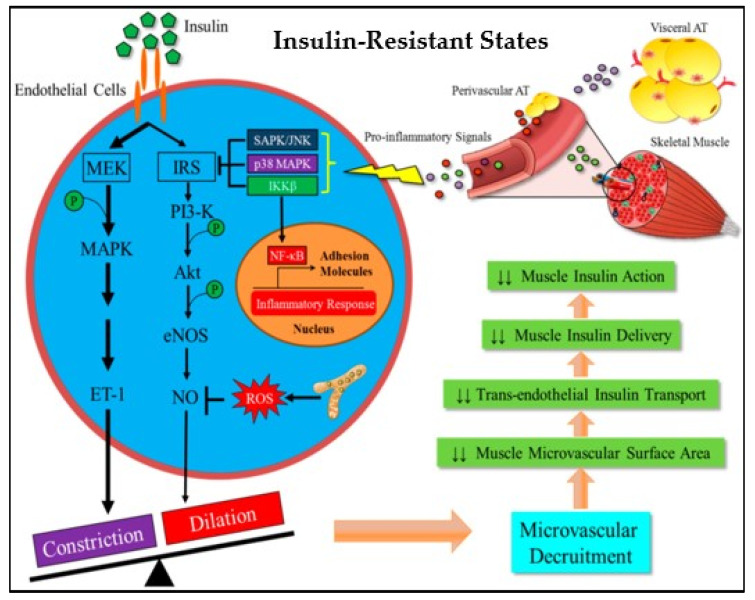
Inflammation in the muscle microvasculature reduces insulin-mediated microvascular recruitment and trans-endothelial insulin transport, adapted from Liu et al., 2019 [39]. MAPK: mitogen-activated protein kinase; ET-1: endothelin-1; NO: nitric oxide; MEK: mitogen-activated protein kinase kinase; IKKβ: nuclear factor-kβ kinase; NF-_K_B: nuclear factor kappa B. ↓↓ = Decreased/Downregulated.

**Figure 5 ijms-25-00469-f005:**
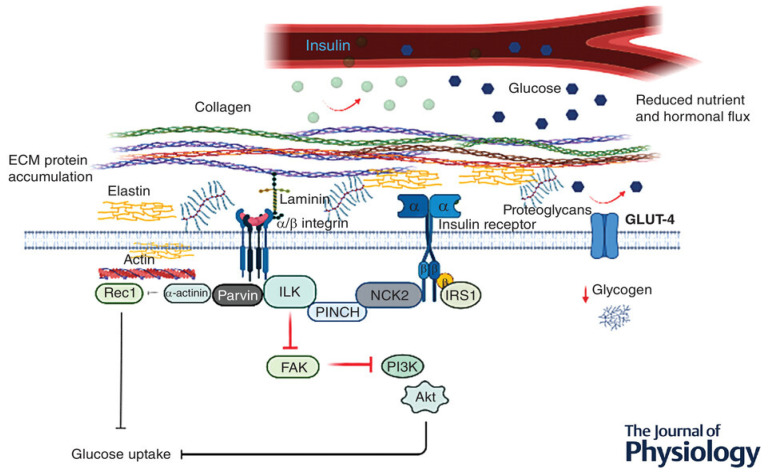
Potential pathway linking integrins and their associated proteins in the regulation of glucose metabolism in skeletal muscle, adapted from Draicchio et al., 2022 [61]. ↓ = Decreased.

**Figure 6 ijms-25-00469-f006:**
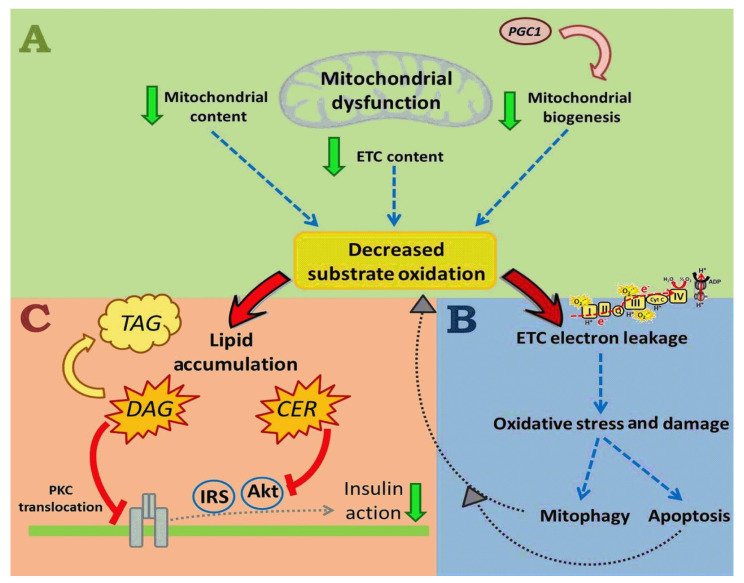
Schematic representation of the pathological mechanism of mitochondrial dysfunction, adapted from Montgomery et al., 2015 [71]. **↓** = Decreased. **↓** = Represents the consequences of mitochondrial dysfunction and electron transport chain (ETC) electron leakage.

**Table 1 ijms-25-00469-t001:** Summarizing the findings of PreDM- and T2DM-associated changes in skeletal muscle function.

PreDM	T2DM
There is an onset of reduced skeletal muscle strength of lower extremities, with hand grip strength being the least affected. Complications are noted over PreDM progression [89,90,94].	Skeletal muscle strength of the lower extremities is prominently affected, with hand grip strength being the least affected. Complications are noted over long-term poorly controlled T2DM [102,103,107,114].
The literature predominantly documents skeletal physical dysfunction in PreDM progression in the context of the effect of PreDM on skeletal muscle strength [89,90,94,95,96].	The type of skeletal muscle fibers and skeletal muscle creatine kinase enzyme are proposed to contribute to skeletal muscle strength and are affected by T2DM [20,128,129,130,135,162].
Mitochondrial dysfunction and skeletal muscle fat infiltration are proposed to be some of the mechanisms behind impaired satellite function in the prediabetic state [68,69,70,77,81].	Skeletal muscle satellite cell function is documented to be impaired by oxidative stress, chronic hyperglycemia, and lipotoxic conditions associated with T2DM [2,135,136,138,139].
The effects of PreDM on skeletal muscle ECM are mainly associated with endothelial dysfunction and skeletal muscle insulin resistance that results from increased skeletal muscle collagen and other ECM proteins. The inflammatory processes associated with PreDM are proposed to contribute to the upregulation of skeletal muscle collagen and ECM proteins [53,57,58,59,62].	T2DM is associated with skeletal muscle ECM remodeling via upregulation of ECM components such as fibronectin, laminin, and collagen, which is proposed to promote thickening of basement membrane and result in diabetes-associated microangiopathy and skeletal muscle insulin resistance. T2D-induced ECM remodeling is observed to affect satellite cell function [54,57,148].

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
