# Peer review of "Prediabetes-Associated Changes in Skeletal Muscle Function and Their Possible Links with Diabetes: A Literature Review"

_ijms, 2023, doi:10.3390/ijms25010469_

Round 1

Reviewer 1 Report

Comments and Suggestions for Authors

The goal of this article is to review the existing literature on diabetic  and prediabetic myopathy  

The aim is interesting and the work is mostly well written but, in my opinion, the paper doesn't reach entirely its goal cause many aspects related to the diabetic myopathy were not considered or not extensively discussed.  Diabetic myopathy is a very complex pathological state and it is reductive to focus only on the aspect concerning satellite cells and muscolar cells development . 

In detail

Section 2. Skeletal muscle progenitor cells and muscle strength. The section talks about satellite cells and musle metabolism but not about muscle strength. Please change the title or the content 

Figure 1. I cannot find corrispondence between the figure and what is written in the text. Please explain better 

Section 4. line 1. The muscle strength depends not only by the number and size of muscle fibres. Many others factors are involved and a section discussing this aspect should be more wider 

Section 4. line 140. The sentence is not correct. The percentage of MHC IIx in adult muscles depends on the species and muscles.  The authors should discuss this aspect or remove the sentence. 

Section 5.3.  A reduction in handgrip strength is not the only complication  of diabetic myopathy and the involved mechanism may be different in different muscle groups. The Authors should broaden the section

Section 6.2 . This section actually talks about the modifications induced by a prediabetic status on ECM proteins and muscle perfusion.  In my opinion the title "effect on skeletal muscle structure" is inappropiate.  

Finally, many references are very old

Reviewer 2 Report

Comments and Suggestions for Authors

The purpose of the manuscript by Dlamini and Khati is to review what is known about alterations in skeletal muscle in prediabetes compared with type 2 diabetes.

General comments:

- Sections 2, 3 and 4 deal mainly with normal physiology, especially satellite cells, myosin expression and muscle strength. Sections 5 and 6 describe some alterations in type 2 diabetes and prediabetes. However, the three parts of the review do not seem to be well connected given the aim of the review. The authors should compare the alterations in prediabetes and diabetes more systematically and address more directly questions such as:

o How do metabolic alterations in prediabetes compare to those in type 2 diabetes?

o How do alterations in satellite cell function or number in prediabetes compare to those in type 2 diabetes?

o How do the ECM alterations in prediabetes compare to those in type 2 diabetes?

o Are alterations in prediabetes and diabetes different or part of a continuum?

- Some of the topics (e.g. MRFs) are prominent in the sections describing normal muscle function, raising the expectation that they will be discussed in detail with respect to (pre)diabetes. However, the subsequent discussion is not very detailed. The manuscript would gain from a better balance and linking of the different parts of the manuscript. The authors should consider shortening the physiology section of the review and providing more detail on alterations in muscle function in type 2 diabetes and prediabetes.

- The authors should consider including tables comparing the main findings on changes in muscle function in type 2 diabetes and prediabetes.

- The findings in humans and animals should be more clearly delineated.

Specific comments:

           “…satellite cells are proposed to play a critical role in muscle fiber maintenance…” Why proposed? They actually play a crucial role.

           “as it is the skeletal muscle that is chiefly involved in regulating glucose uptake and maintaining glucose homeostasis” Perhaps the role of the liver should also be mentioned. The liver also plays an important role.

- Lines 65-70: The fraction of glucose taken up by skeletal muscle depends on the route of glucose delivery (oral or intravenous). After a meal, much of the glucose ends up in the liver, so the muscles may take up less than 80% of it.

- The original paper in which the satellite cells were described is not cited.

- The difference between human and rodent muscles in terms of the presence of type IIx and IIb fibers should be more clearly elaborated.

- Lines 240-245: did high levels of TNF-alpha cause a decrease in muscle mass or were they just associated/correlated with loss of muscle mass?

Round 2

Reviewer 1 Report

Comments and Suggestions for Authors

The paper sounds better even if the contribute in the field  of Diabetic myopathy, in my opinion, remains not very high

Introduction. Move the sentence:  "Alteration to.... [2,4]" (line 30-32)  after [1] 

Line 55. it has already been said. Please remove the sentence 

Chapter 2: the chapter talks about satellite cells so please remove  "The skeletal muscle to ....[4]

Chapter 3: it is not ncessary to explain what are the satellite cells . Remove line 77 and 78

Again I cannot find correspondence between the figure 1 and what is written in the text.  I cannot find in the figure MEF-1 for example 

Reviewer 2 Report

Comments and Suggestions for Authors

Comments:

- The changes in muscle function in prediabetes and their mechanistic links to type 2 diabetes are still not clearly described (see my original comments).

-PD is not the optimal abbreviation for prediabetes as it is often used for Parkinson’s disease.

-The abstract states: “Hence, research on PD is essential because establishing diabetic biomarkers during the prediabetic state would aid in preventing the development of T2DM, as PD is a reversible condition if it is detected in the early stages.« However, the review does not really address markers for the prediabetic state.

-Lines 70-72: Translocation of GLUT2 is not a mechanism for postprandial glucose disposal.

-Lines 450-452: What is meant by amelioration of satellite cell function?

-Lines 415 and 416: Inhibition of Akt is not the same as inhibition of the insulin receptor.

-“The loss of skeletal muscle mass and strength caused by T2DM results in decreased surface area for glucose transport, further exacerbating insulin resistance” How does loss of strength decrease the surface area?

-Figures such as Figure 2, Figure 3, Figure 4, Figure 5, Figure 6… do not appear to be original as they are (too) similar to those in the cited literature.

-“Type I levels directly correlate with insulin sensitivity; hence the insulin resistance associated with T2D considerably affects the expression of Type I fibers [58]” Does insulin resistance reduce the proportion of type I fibres or does insulin resistance result from a reduced proportion of type I fibres?

“Integrin, a crucial receptor of ECM molecules, influences the regulation insulin activity.” Which integrin?

-The role of LPL should be described more clearly.

-“In another study, inhibition of C2C12 myoblast differentiation was observed in palmitic acid-induced insulin resistance model, whereas glucose uptake activity and insulin sensitivity gradually increased during C2C12 myoblast differentiation [136]. Thus, these findings confirm that prediabetes can inhibit myoblast differentiation and decrease skeletal muscle function [119].” C2C12 cells are grown in vitro, whereas prediabetes is diagnosed in vivo. Findings in vitro therefore cannot confirm what is going on in the prediabetic state.

-“Insulin resistance induces impaired fasting glucose levels, increased plasma glucose, and elevated FFA concentrations, inhibiting myoblast differentiation” What is the difference between impaired fasting glucose levels and increased plasma glucose levels?

-“ The increased inflammatory cytokines during the prediabetic state, suggested to be induced by inter- and intramuscular adipose tissue accumulation, are documented to be associated with reduced muscle mass and strength [53].” It is not clear what is meant by intermuscular adipose tissue. Did the authors mean to refer to adipose tissue or fat accumulation?

-“ Protein Kinase C activates DAG, promoting the translocation of DAG to the plasma membrane, which inhibits the insulin receptor [121].” This description is not correct.

-“ The reduction in skeletal muscle fiber MHC and MLC isoforms results in diminished muscle fiber number and size, thereby affecting muscle mass and muscle strength.” Is the reduced expression of MHC and MLC the cause or a marker for the loss of muscle fibres?

-Statements such as “In another study, biopsies from T2DM patients illustrated a reduced oxidative metabolism program, along with increased type 2x fibers [59]” are not very informative. What does oxidative programme mean? What does an increase in 2x fibres mean (increased number, increased size,…)?

-Abbreviations should be explained. For example, MHC and MLC are used without any explanation.

-“ Studies document that insulin induces vasodilation in resistant arterioles, increases compliance of large arteries, promotes capillary recruitment, and maintains capillary permeability to support nutrient delivery [89, 90].” Did the authors mean that arterioles are resistant to insulin, or did they have the resistance arterioles in mind?

- Lines 346-347: “Current literature mainly focuses on microvascular and macrovascular prediabetic complications such as endothelial dysfunction and cardiovascular disease.” It is not clear which complications the authors have in mind here.

-In line 270, the authors say that prediabetes is an asymptomatic condition. However, if it leads to complications, it is not asymptomatic.

-“ Understanding the modifications that occur in the skeletal muscle during the prediabetic state can allow us to be able to target and prevent the processes that contribute toward the development of diabetic myopathy in the prediabetic state” Can diabetic myopathy develop when the patient does not have diabetes (but has prediabetes)?

Comments on the Quality of English Language

No comments.
